# Effectiveness of time-varying echo information for target geometry identification in bat-inspired human echolocation

Miwa Sumiya[1,2]*, Kaoru Ashihara[3], Hiroki Watanabe[4], Tsutomu Terada[5], Shizuko Hiryu[6], Hiroshi Ando[1]

1 Center for Information and Neural Networks (CiNet), National Institute of Information and Communications Technology (NICT), Seika-cho, Kyoto, Japan, 2 Research Fellow of Japan Society for the Promotion of Science, Chiyoda-ku, Tokyo, Japan, 3 Human Informatics and Interaction Research Institute, National Institute of Advanced Industrial Science and Technology, Tsukuba, Ibaraki, Japan, 4 Faculty of Information Science and Technology, Hokkaido University, Sapporo, Hokkaido, Japan, 5 Graduate School of Engineering, Kobe University, Kobe, Hyogo, Japan, 6 Faculty of Life and Medical Sciences, Doshisha University, Kyotanabe, Kyoto, Japan

* miwa1804@gmail.com

**Data Availability Statement:** All relevant data are within the paper and its Supporting Information files.

## Abstract

Bats use echolocation through flexible active sensing via ultrasounds to identify environments suitable for their habitat and foraging. Mimicking the sensing strategies of bats for echolocation, this study examined how humans acquire new acoustic-sensing abilities, and proposes effective strategies for humans. A target geometry identification experiment—involving 15 sighted people without experience of echolocation—was conducted using two targets with different geometries, based on a new sensing system. Broadband frequency-modulated pulses with short inter-pulse intervals (16 ms) were used as a synthetic echolocation signal. Such pulses mimic buzz signals emitted by bats for echolocation prior to capturing their prey. The study participants emitted the signal from a loudspeaker by tapping on Android devices. Because the signal included high-frequency signals up to 41 kHz, the emitted signal and echoes from a stationary or rotating target were recorded using a 1/7-scaled miniature dummy head. Binaural sounds, whose pitch was down-converted, were presented through headphones. This way, time-varying echo information was made available as an acoustic cue for target geometry identification under a rotating condition, as opposed to a stationary one. In both trials, with (i.e., training trials) and without (i.e., test trials) answer feedback immediately after the participants answered, the participants identified the geometries under the rotating condition. Majority of the participants reported using time-varying patterns in terms of echo intensity, timbre, and/or pitch under the rotating condition. The results suggest that using time-varying patterns in echo intensity, timbre, and/or pitch enables humans to identify target geometries. However, performance significantly differed by condition (i.e., stationary vs. rotating) only in the test trials. This difference suggests that time-varying echo information is effective for identifying target geometry through human echolocation especially when echolocators are unable to obtain answer feedback during sensing.

**Funding:** This study was supported by JSPS KAKENHI Grant Number JP18J01429 (Grant-in-Aid for JSPS Fellows) to MS. The funders had no role in study design, data collection and analysis, decision to publish, or preparation of the manuscript.

**Competing interests:** The authors have declared that no competing interests exist.

## Introduction

Echolocating bats recognize their environments by emitting ultrasounds and listening to echoes from objects in a process called echolocation. Bats have adapted such echolocation signals to locate their habitat and prey [1, 2]. Moreover, bats change the acoustic features of echolocation signals (e.g., frequency, duration, and sound pressure) and strategies for signal emissions (e.g., timing, rate, and direction) according to their flight environment and purpose [3–9]. Such flexible acoustic sensing using echolocation signals enables advanced sonar behaviors among animals [10, 11]. In the same manner, humans can use echolocation to navigate their environment [12, 13]. Previous studies have measured advanced echolocation performance, such as distance discrimination [14], among blind echolocation experts who use self-produced mouth clicks. Such experts can identify two-dimensional (2D) shapes by moving their heads while sensing through mouth clicks [15], and can adjust the number and intensity of clicks to detect a target presented from distant angles, thereby compensating for weak echo intensities [16]. Evaluating and modeling sensing strategies are not only scientifically interesting but also useful for instructing and guiding new users of human echolocation.

In this light, this study examined how humans can acquire new acoustic-sensing abilities by mimicking the strategies used by bats, and proposes effective strategies for humans. In a previous study [17] (hereinafter, the previous study) conducted by the authors of the present study, a psychoacoustic experimental system was constructed using binaural echoes, which were recorded using synthetic echolocation signals. The previous study examined the applicability of a bat-inspired signal design in human echolocation. The pitch of binaural echoes was down-converted after a miniature dummy head (MDH) [18] captured echoes, because there were high-frequency signals of up to 35 kHz. The pitch-converted binaural echoes were then presented to the participants of the previous study. The pitch-conversion rate matched the scale of the MDH to compensate for the scale difference between the human head and MDH. This suggests that listening to pitch-converted binaural echoes provides listeners with the sensation of listening to real spatial sounds in a three-dimensional (3D) space. In the previous study, the participants without vision loss distinguished texture by listening to the timbre of echoes; this suggests that texture can be distinguished by using synthetic echolocation signals with suitable frequency bands and time–frequency structures [17]. However, the previous experiment on distinguishing texture was performed using a three-interval, two-alternative forced-choice task, which implies that the participants distinguished texture by simply determining whether the first and second (or second and third) sound stimuli differed. Therefore, it remains unclear whether humans can also "identify" texture simply by the design of synthetic echolocation signals.

In bat biosonar research, some studies suggest that echolocating bats that use frequency-modulated (FM) signals (FM bats) recognize the parts of plants (i.e., flowers) and distinguish fine surface texture using echo information [19, 20]. For example, nectar-feeding bats (*Glossophaga commissarisi*) determine flowering stages through echolocation to identify nectar-rich flowers [19]. In addition, the same study suggests that FM bats identify a flower's specific shape and texture by listening to spectral directional echo patterns. This implies that FM bats may use angular-dependent, time-varying echo information to recognize flowers. Moreover, Falk *et al*. [20] find that another species (*Eptesicus fuscus*) can distinguish smooth objects from size-matched textured objects while flying. Furthermore, the present study proposes that FM bats listen to changes in the sound spectra of consecutive echoes to distinguish texture. This means that they may use time-varying echo information given by changes in the relative positions of objects while flying to distinguish texture. In summary, previous studies suggest that FM bats use "time-varying echo information" for object recognition and texture

discrimination by emitting echolocation signals repeatedly toward objects from various angles and positions and listening to time-varying echoes. Moreover, another study on human echolocation demonstrates that blind echolocation experts can identify 2D shapes by sensing targets from various angles using self-produced sounds, such as mouth clicks, while moving their heads [15]. This suggests that similar to FM bats, humans may use the time-varying echo information they acquire to sense objects from different angles and positions, especially when they cannot obtain reliable target information from echoes acquired from a single angle.

In light of the discussion above, the present study investigated whether time-varying echo information can be used effectively to identify target geometry through human echolocation. An experiment in this regard was constructed on the basis of a proposed sensing system, using two targets with different geometries (i.e., cylinders with patterned indented surfaces). The study recruited 15 participants who were tasked to tap Android devices to generate a synthetic echolocation signal emitted from a loudspeaker. Because the signal included high-frequency signals up to 41 kHz, the signal and echoes from the target were recorded using an MDH, which was also used in the previous study [17]. The participants identified the target presented in front of a loudspeaker by listening to pitch-converted binaural sounds. The performances under two experimental conditions with (rotating condition) and without (stationary condition) time-varying echo information as acoustic cues were compared. Under the rotating condition, the target rotates in front of the loudspeaker to simulate time-varying echo information. This condition mimics the situation where FM bats emit ultrasounds toward objects from different angles as they fly to listen to time-varying echoes. The synthetic echolocation signal consists of broadband FM pulses with short inter-pulse intervals (IPI). The signal mimics the buzz signal emitted by FM bats at the terminal phase, before they capture their prey [2]. As its main contribution, this study is the first to examine effective sensing strategies for target geometry identification through human echolocation, following the strategies of echolocating bats.

## Materials and methods

### Sensing system

Fig 1 illustrates the sensing system, which comprises three parts: 1) emission of a synthetic echolocation signal (Fig 1A, red lines), 2) binaural recording of the emitted signal and its echoes from the target using a 1/7-scaled MDH (Fig 1A, blue lines; Fig 1B), and 3) pitch conversion of recorded binaural sounds and presentation of pitch-converted binaural sounds to the participants (Fig 1A, green lines). The system is operational once the participants tap the gray buttons on the application screens of two Android devices (Fig 1C).

**Synthetic echolocation signal.**   Downward linear FM pulses (pulse duration: 3 ms; frequency: 41–7 kHz; number of pulses: 325 pulses; sampling frequency: 96 kHz) with short IPIs (16 ms) were used as the synthetic echolocation signal (Fig 2). The signal mimicked ultrasonic pulses with short IPIs emitted by echolocating bats at the terminal phase (also known as buzz) before capturing their prey [2]. The frequency band of the signal was determined by the scales of the targets and the MDH, considering the intensity–frequency characteristics of the human auditory system [21]. The frequency response of the signal was digitally corrected before conducting the experiment to equalize the frequency response of the system. After this correction, an experimenter confirmed that the frequency response of the overall system was within ± 3 dB within a range of 8 kHz to 40 kHz.

**System configuration.**   Android device A (MediaPad M5 SHT-W09; Huawei Technologies Co., Ltd., Shenzhen, China) with a 16-bit accuracy at a sampling rate of 96 kHz digitally generated a synthetic echolocation signal, which was converted to an analog signal using audio

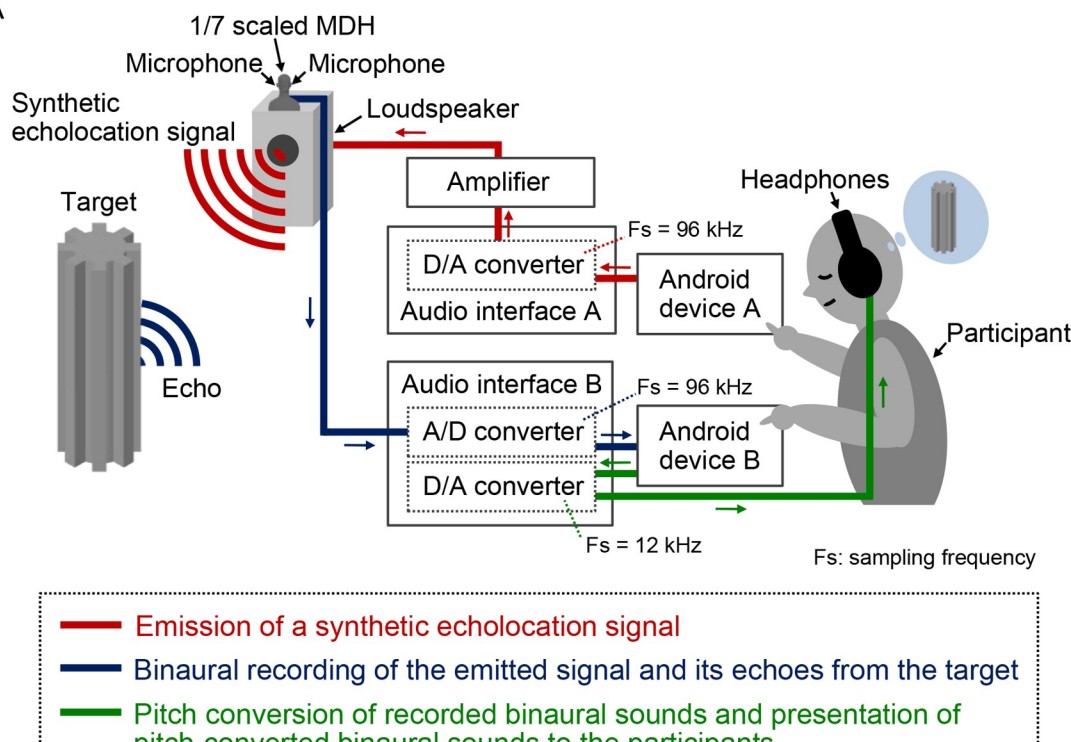

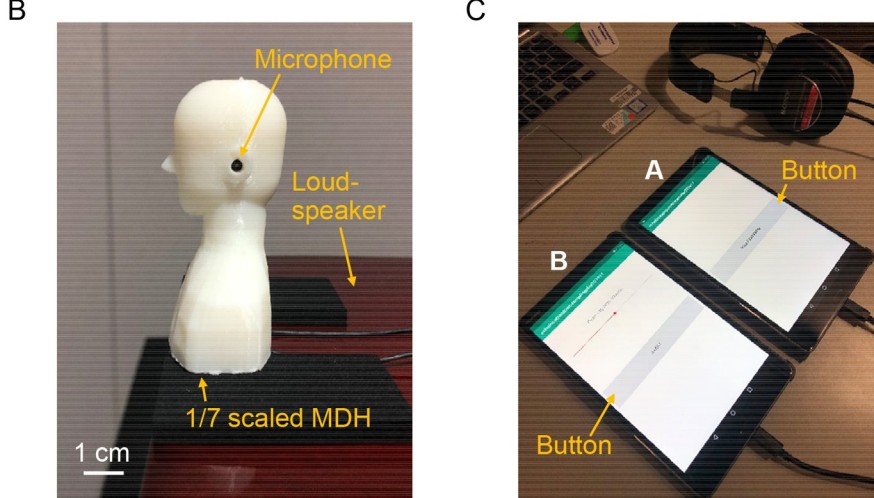

**Fig 1. Sensing system.** (A) denotes the configuration of the sensing system for target geometry identification. Once the participants tap Android devices A and B, the synthetic echolocation signal is emitted from the loudspeaker (red lines). The emitted signal and its echoes from the target are recorded using the 1/7-scaled MDH (blue lines). The recorded binaural sounds, whose pitch is converted to 1/8 of the original by lowering the sampling frequency, are presented to the participants through headphones (green lines). (B) denotes the 1/7-scaled MDH with two microphones inserted in its ear canals for the binaural recordings. (C) denotes the Android devices A and B.

interface A (U-22; Zoom Corporation, Tokyo, Japan). The signal processed by an audio amplifier (Olasonic NANO-A1; Inter Action Corporation, Kanagawa, Japan) was emitted from a loudspeaker (S-300HR; TEAC Corporation, Tokyo, Japan). The sound pressure level ($L_p$) of

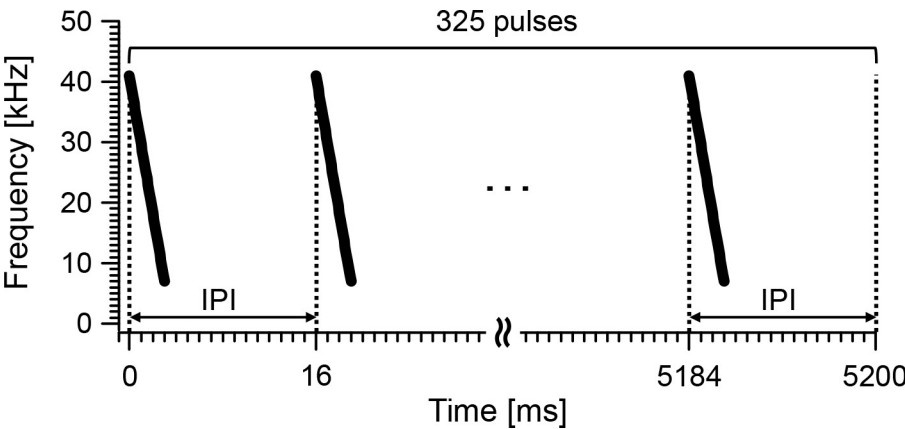

**Fig 2. Synthetic echolocation signal.** Downward linear FM pulses (pulse duration: 3 ms; frequency: 41–7 kHz; number of pulses: 325 pulses; sampling frequency: 96 kHz) with short IPIs (16 ms) were used as the synthetic echolocation signal; this mimicked buzz signals emitted by bats before capturing their prey.

the signal was 117 dB and was situated 10 cm away from the loudspeaker. The emitted signal (i.e., sound directly from the loudspeaker) and its echoes from the target were recorded using two omnidirectional condenser microphones (B6 Omnidirectional Lavalier; Countryman Associates, Inc., CA, USA) inserted 3 mm from the entrance of the left and right ear canals of the 1/7-scaled MDH (Fig 1B). Audio interface B (U-24; Zoom Corporation) was used to digitalize the binaural sounds with a 16-bit accuracy at a sampling rate of 96 kHz and to store the digitalized sounds synchronously on Android device B (MediaPad M5 SHT-W09). The pitch of the recorded binaural sounds was converted to 1/8 of the original by lowering the sampling frequency from 96 kHz to 12 kHz (pulse duration: 24 ms; frequency: 5.125–0.875 kHz; IPI: 128 ms; sampling frequency: 12 kHz). The pitch conversion rate used was 1/8 instead of 1/7 to round off the number from 13,714 to 12,000 for the sample rate. The pitch-converted binaural sounds were presented to the participants via headphones (MDR-CD900ST; Sony Corporation, Tokyo, Japan) at the entrance of the right ear canal, at sound pressure levels ($L_p$) ranging from 56 dB to 69 dB. The sensing system was constructed through Android applications based on EMUI version 9.0.1, a mobile operating system.

The sensing system employed two Android devices and audio interfaces to avoid unwanted sounds through the headphones and loudspeaker. If the system used a single Android device and a single audio interface, the synthetic echolocation signal would play automatically, not only on the loudspeaker but also on the headphones, before presentation of the pitch-converted binaural sounds. Moreover, the same scenario would occur for pitch-converted binaural sounds.

## Experiment

**Participants.** The study recruited 15 sighted participants designated as P1 to P15 (males = 8), respectively. Their ages ranged from 22 years to 39 years (mean ± standard deviation: 30.4 ± 6.2 years). The experiment was conducted at the National Institute of Information and Communications Technology (NICT) in Japan. Prior to the experiment, a hearing test was conducted for each participant using an audiometer (AA-58; RION Co., Ltd., Tokyo, Japan) with tone bursts of 250, 500, 1,000, 2,000, 4,000, and 8,000 Hz in a sound-attenuated chamber (HLW: 2.3 m × 1.8 m × 2.0 m) to verify that the participants possessed normal hearing range. The pure tone thresholds in the left and right ears of the participants were within a hearing level of 30 dB from 250 Hz to 8,000 Hz, except for P4, who exhibited a pure tone

threshold of 35 dB at 8,000 Hz in the right ear (S1 Fig). Nevertheless, the data for P4 were included in the analysis because the pitch-converted binaural sounds presented in the experiment were not above 8 kHz (approximately 1–5 kHz). All of the participants reported no experience of echolocation. The Ethics Board of the NICT approved the psychoacoustic experiments. All of the participants provided written informed consent before participating in the hearing test and the experiment.

**Targets.** Fig 3 displays the targets (1 and 2) used in the experiment. The targets were cylinders with different geometries and were printed using a 3D printer (Replicator+; MakerBot, NY, USA) with polylactic resin. The convex sides were attached equi-angularly on the surfaces of the targets, whereas the pitch of the convex sides of target 1 was two times longer than those of target 2. The targets were 8 cm in diameter and 36 cm in length.

**Experimental system and general task.** The experiment was conducted inside two equal-sized sound-attenuated chambers (HLW: 2.3 m × 1.8 m × 2.0 m; Fig 4). The participants wore headphones and remotely sensed the targets by tapping the two Android devices in chamber 1. The participants in chamber 1 could not view chamber 2, which was blocked by walls. The target was placed in chamber 2 such that the $Y$ and $Z$ coordinates between the sound source and the center of gravity of the target (Fig 4, left, $g$ marks) matched in the $YZ$ plane. The distance between the target and loudspeaker was 23.8 cm. In addition, the MDH was situated such that the $Y$ coordinates between the sound source and the center of gravity of the MDH matched. Background noise levels ($L_A$) in chambers 1 and 2 were approximately 17 dB (chamber 1) and 18 dB (chamber 2), respectively, which were sufficiently low to prevent intervention with the pitch-converted binaural sounds by using headphones (chamber 1) and to enable measurement of echoes from the target (chamber 2).

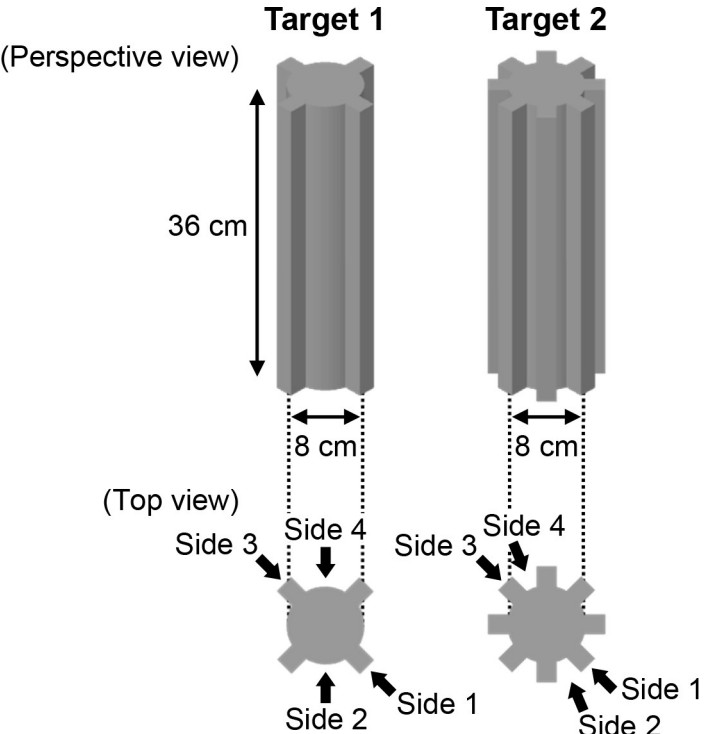

**Fig 3. Targets.** This figure shows the perspective (top) and top (bottom) views of targets 1 and 2 that were used in the experiment. Sides 1 to 4 of the targets were used to place the targets in the experiment (for details, see "Procedure" under the "Experiment" section).

(Top view)

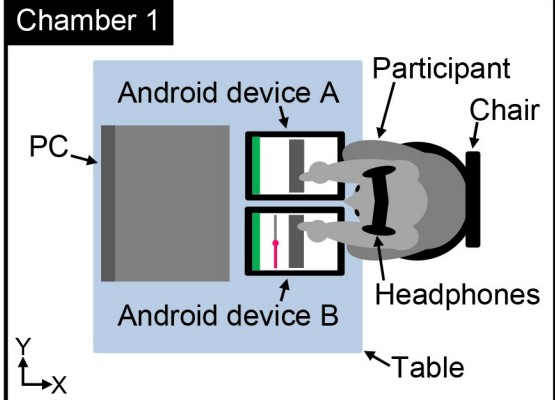

(Side view)

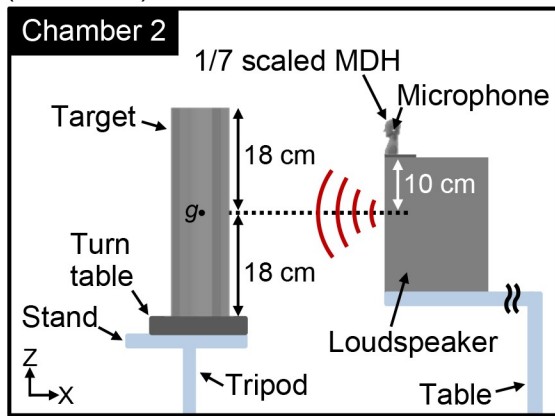
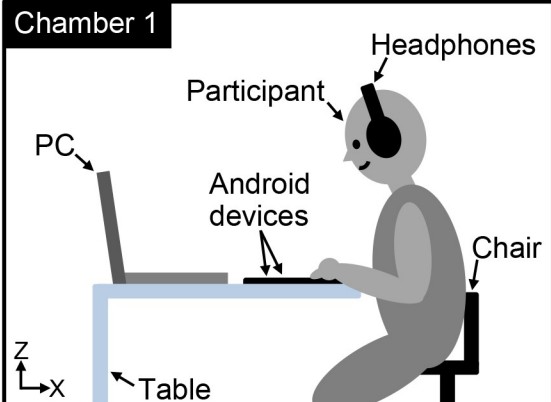

**Fig 4. Experimental system.** This figure shows the top and side views of the experimental system. Once the participants tap Android devices A and B in chamber 1, the synthetic echolocation signal is emitted from the loudspeaker to the target (target 1 or 2) in chamber 2. The emitted signal and its echoes from the target are then recorded using the 1/7-scaled MDH on the loudspeaker. The recorded binaural sounds, whose pitch is converted to 1/8 of the original, are presented to the participants through the headphones in chamber 1. Then the participants identify the target geometries by listening to the pitch-converted binaural sounds by pressing the keys of the notebook computer in chamber 1. Two experimental conditions are employed, namely, rotating (i.e., the target is rotating horizontally on an electric turntable) and stationary (i.e., the target remains stationary on the electric turntable). The red arrow in the top view indicates the direction of the rotation of the target.

The participants initiated sensing by tapping the Android devices once after the notebook computer (Inspiron 7380; Dell Inc., TX, USA) placed in chamber 1 displayed the instruction to begin sensing. The participants tapped device A for pulse emission (5.2 s) immediately after tapping device B to listen to all pitch-converted binaural sounds because device B employs a limit of 6.052 s per instance of sensing during recording. The participants were instructed to refrain from adjusting the volume of the pitch-converted binaural sounds, as the sounds were confirmed to be within the acceptable range. After listening to the pitch-converted binaural sounds, the participants determined whether target 1 or 2 was placed in front of the loudspeaker in chamber 2. The participants input their responses by pressing the keys of the notebook computer immediately after presentation of the instruction for answering. An experiment control software (Presentation; Neurobehavioral Systems, Inc., CA, USA) installed on the notebook computer was used to present the instructions and record the participants' responses.

**Experimental conditions.** Experimental conditions were employed to define whether the target was rotating or stationary. Under the rotating condition, the target rotated horizontally

with the electric turntable at a rate of approximately 2.95 rpm (0.28° between pulses) in a counterclockwise direction (Fig 4, red arrow). By rotating the target instead of moving it around while sensing, the experiment mimicked echolocating FM bats as they emit ultra-sounds toward objects from various angles while flying to listen to time-varying echoes. The synthetic echolocation signal was emitted for 5.2 s from the loudspeaker for each instance of sensing. Thus, the participants could listen to the pitch-converted binaural echoes from approximately 1/4 round of the target (92°) for 41.6 s (corresponding to eight times longer than the signal duration of 5.2 s) under the rotating condition. However, under the stationary condition, the participants could not sense the target from different angles because it was stationary on the electric turntable. Therefore, time-varying echo information was available as the acoustic cue for target geometry identification under the rotating but not under the stationary condition.

Although the background noise level in chamber 2 under the rotating condition was approximately 1 dB higher ($L_A$ = 19 dB) than that under the stationary condition ($L_A$ = 18 dB) because of the use of the electric turntable, measuring echoes from the rotating target remained acceptable. P1 to P8 participated in the rotating and stationary conditions on the first day and 1 week after (experimental order A), respectively. Meanwhile, P9 to P15 participated in the stationary and rotating conditions on the first day and 1 week after (experimental order B), respectively. Different orders were employed to verify order effect (for details, see "Statistics" under the "Experiment" section).

**Procedure.** Under each condition, training trials (with answer feedback) were followed by test trials (without answer feedback). The answer feedback from the training trials was presented as 3D graphics of the targets using the experiment control software on the display of the notebook computer, which was presented to the participants immediately after their responses. The participants were not given any information on acoustic cues prior to the experiment. This means that they were trained independently by verifying the answer feedback during the training trials. The experimenter stayed in chamber 2 to monitor remotely the display of the notebook computer in chamber 1 and to place the targets on the stationary or rotating turntable in chamber 2, accordingly. Under the stationary condition, side 1 or 2 (1 to 4) of the targets randomly faced the loudspeaker in the training (test) trials (Fig 3). In the rotating condition, the target was placed on the electric turntable 5 s before the instruction to start sensing. Therefore, the target rotated upon placement on the turntable instead of upon emission of the sound. At placement, side 1 or 2 (1 to 4) of the targets randomly faced the loudspeaker in the training (test) trials (Fig 3). Sensing was initiated by tapping the Android devices. Thus, the participants could not hear the same echoes under the rotating condition. This way, the study prevented familiarization of the participants with the echoes associated with the target geometries during the training trials under the rotating condition. A total of 16 training trials (2 targets × 2 presentation patterns × 4 repetitions) and 16 test trials (2 targets × 4 presentation patterns × 2 repetitions) were performed under both conditions. The training and test trials were conducted in four sessions with four trials per session and 1–2 min breaks between sessions. In the case where the pulses and echoes were recorded at less than 4/5 of the sensing duration due to a delay in tapping device A for pulse emission, the response was excluded from the experimental data to ensure data reliability. Specifically, five out of 480 trials (32 trials × 15 participants) and three out of 480 trials (32 trials × 15 participants) were excluded under the stationary and rotating conditions, respectively.

Before the experiment, the experimenter briefed the participants about the procedure. During the briefing, the two actual targets were shown to allow the participants to recognize visually the 3D shapes and target geometries. To avoid confusion, the participants were not informed that the pitch of the sounds presented through the headphones was converted.

Moreover, the participants were not informed about the distance between the loudspeaker and the target. Thus, the study assumed that no confusion was introduced about the scale differences between the visual and acoustic images of the targets, although the participants listened to pitch-converted binaural sounds. The participants under the rotating condition were instructed that the target will be rotating horizontally at a constant rate and that they will be listening to the echoes from approximately 1/4 round of the target while sensing. Meanwhile, under the stationary condition, the participants were instructed that the target was stationary while sensing. However, under both conditions, they were not instructed as to which sides (stationary condition) and parts (rotating condition) were facing the loudspeaker.

**Hearing survey.** In a study on dolphin biosonar, DeLong *et al.* [22] conducted a human listening experiment using pitch-converted echoes recorded from a typical bottlenose dolphin click to detect the acoustic features used by dolphins in distinguishing material and wall thickness. Moreover, DeLong *et al.* [22] conducted a hearing survey on acoustic cues to take advantage of the fact that human subjects can verbally report discriminatory cues. Building on DeLong *et al.* [22], the previous study [17] conducted a hearing survey for sighted participants to examine the effectiveness of the acoustic cues used in distinguishing shape, texture, and material through synthetic echolocation signals. In the present study, a hearing survey was also conducted to examine the effectiveness of the acoustic cues used by each participant for target geometry identification. The participants were encouraged to describe in their own words the acoustic cues they used to identify the target geometries. The answers were categorized as time-varying or non-time-varying echo information. The study predicted that time-varying and non-time-varying echo information will be used in the rotating and stationary conditions, respectively. Moreover, the specific acoustic features of time-varying or non-time-varying echo information that participants used as acoustic cues were categorized into four groups, namely, (1) intensity, (2) timbre, (3) pitch, and (4) others.

**Statistics.** To verify whether the experimental orders (for details, see the "Experimental conditions" section) influenced target geometry identification, generalized linear mixed models (GLMMs) [23] were built using binomial distribution with logit link. In GLMMs, the number of correct answers relative to the number of false answers for each target was considered as the response variable, whereas the experimental order was designated as the factor-type explanatory variable. The participant's ID was treated as the random effect. The GLMM evaluations indicate that performance was not influenced by the experimental order (experimental order A [stationary condition]: $\beta = 0.276 \pm 0.397$, $z = 0.696$, $p = 0.487$ [S2 Fig, left panel]; experimental order B [rotating condition]: $\beta = -0.391 \pm 0.379$, $z = -1.030$, $p = 0.303$ [S2 Fig, right panel]). Therefore, the performance of all participants was included in subsequent data analyses.

To evaluate the influence of the experimental conditions (stationary and rotating) on target geometry identification, other GLMMs were built using binomial distribution with logit link. The number of correct responses relative to the number of false answers for each target was considered as the response variable, whereas the experimental condition was assigned as the factor-type explanatory variable. In addition, a chance condition was introduced and considered as the factor-type explanatory variable to compare the actual performance with chance. As the chance level in the experiment was 50%, the number of correct responses under the chance condition was assumed to be half of the number of the trials for each target (i.e., four answers out of eight were assumed correct for each target). The participant's ID was treated as the random effect.

We used the statistical computing environment R (version 3.6.1) for GLMM analyses. All parameter estimations for GLMMs were conducted using the "glmmML" function (package "glmmML" version 1.1.0) [24] and the Gauss–Hermite quadrature method. MATLAB (The

MathWorks, Inc., MA, USA) was used to visualize the target geometry identification performance.

## Results

### Target geometry identification performance

Fig 5A and 5B present the target geometry identification performance of the participants and the results of the GLMM evaluation, respectively. Although majority of the participants (12/15) in the training trials displayed performance above the chance level under the stationary and rotating conditions (Fig 5A, left panel), they exhibited extremely high performance only under the rotating condition (P1: 81.3%; P12: 87.5%). The GLMM evaluations indicate that the participants in the training trials were more likely to identify the target geometries correctly than chance performance under the rotating condition (Fig 5B, middle left panel; $\beta = 0.533 \pm 0.186$, $z = 2.863$, $p < 0.01$) but not under the stationary condition (Fig 5B, top left panel; $\beta = 0.360 \pm 0.185$, $z = 1.946$, $p = 0.052$). Furthermore, the GLMM evaluation indicates that the participants in the training trials were not more likely to identify the target geometries correctly under the rotating condition than under the stationary condition (Fig 5B, bottom left panel; $\beta = 0.174 \pm 0.189$, $z = 0.921$, $p = 0.357$).

Although the target geometry identification performance of the participants in the test trials displayed more variations than those in the training trials under the stationary and rotating conditions (Fig 5A, right panel), the median of the proportions of correct responses increased only under the rotating condition [stationary: 56.3% (training), 56.3% (test); rotating: 62.5% (training), 68.8% (test)]. Under the stationary condition, the number of participants with performance less than or equal to the chance level increased in the test trials (training: 3/15 participants; test: 6/15 participants), and the test performances of P2 (25.0%), P8 (18.8%), P10 (37.5%), and P15 (18.8%) were extremely low. Under the rotating condition, the number of participants with extremely high performance (above 80%) increased in the test trials (training: 2/15 participants; test: 4/15 participants), and P1 and P12 displayed the highest test performance (93.8%); they also showed extremely high performance in the training trials. The GLMM evaluations indicate that the participants in the test trials were more likely to identify the geometries correctly than chance performance under the rotating condition (Fig 5B, middle right panel; $\beta = 0.750 \pm 0.189$, $z = 3.964$, $p < 0.001$) but not under the stationary condition (Fig 5B, top right panel; $\beta = 0.076 \pm 0.183$, $z = 0.413$, $p = 0.680$). Moreover, the GLMM evaluation indicates that the participants in the test trials were more likely to identify the geometries correctly under the rotating condition than under the stationary condition (Fig 5B, bottom right panel; $\beta = 0.700 \pm 0.193$, $z = 3.619$, $p < 0.001$).

### Acoustic cues for target geometry identification

The previously described synthetic echolocation signal was used to distinguish successfully the differences in the time-varying patterns of echo amplitudes (Fig 6, top panels) and spectrograms (Fig 6, bottom panels) between the two targets under the rotating condition. The time-varying echo patterns in target 2 changed more rapidly than those in target 1 because target 2 had twice the number of target 1's convex sides. The time-varying patterns in the echo spectrograms remarkably appeared in the pitch-converted frequency range of 2–4 kHz (corresponding to the actual frequency range of 16–32 kHz) for targets 1 and 2. Although the sound pressure levels ($L_p$) of the pitch-converted binaural sounds (at the entrance of the right ear canal) for target 1 (maximum: $67.2 \pm 0.06$ dB [mean ± standard error]; minimum: $58.5 \pm 0.04$ dB; mean: $63.1 \pm 0.04$ dB; $N = 120$ [8 trials × 15 participants]) were higher than those for target

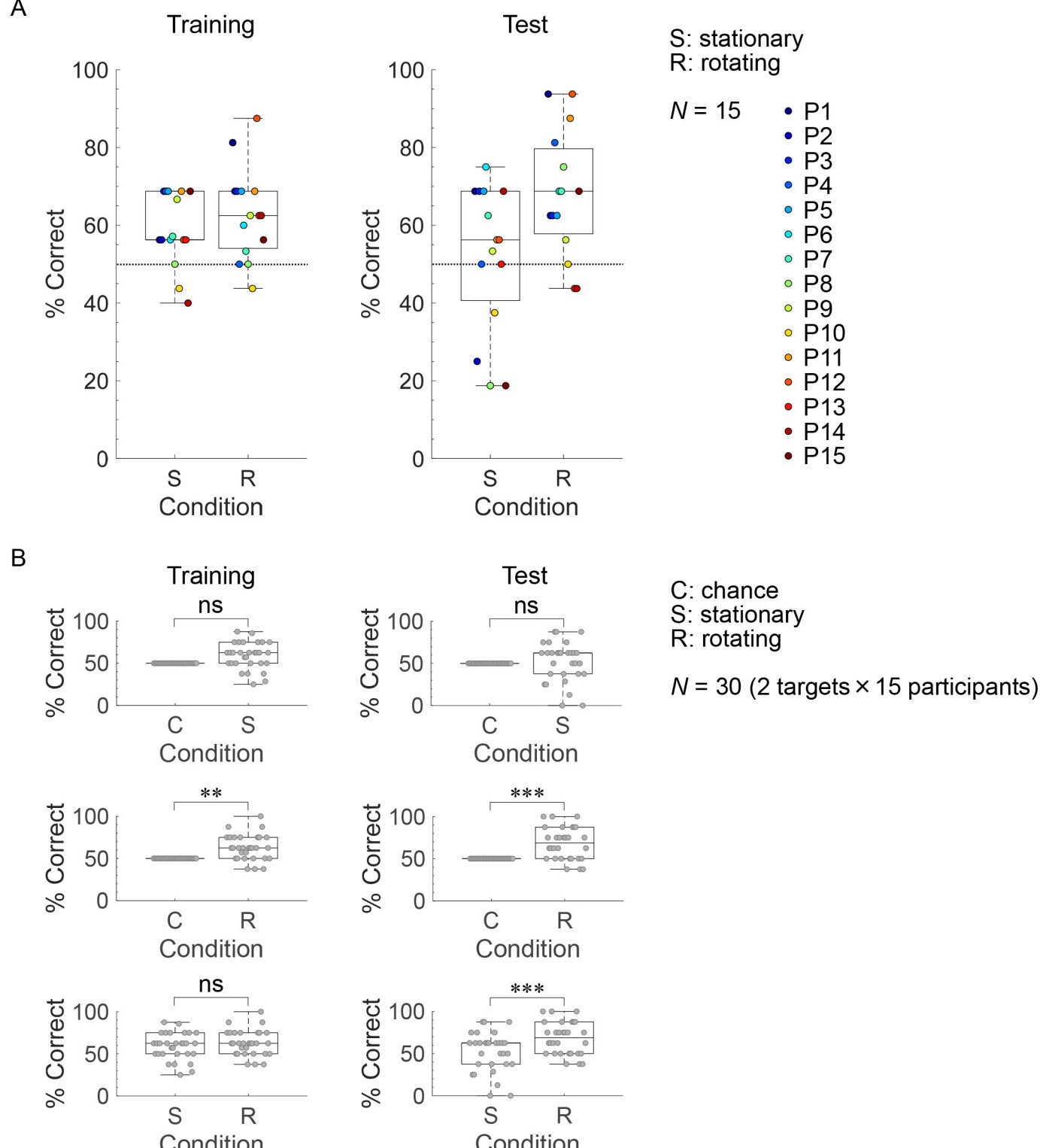

**Fig 5. Target geometry identification performance.** (A) shows box-and-whisker and scatter plots of the proportions of correct responses (P1–15) in the training (left) and test (right) trials under the stationary (S) and rotating (R) conditions. The horizontal lines in the boxes indicate the median. The bottom and top of the boxes indicate the 25th ($q_1$) and 75th ($q_3$) percentiles, respectively. The bottom and top whiskers extend to the minimum and maximum data points within the range of $[q_1 - 1.5 \times (q_3 - q_1)]$ to $[q_3 + 1.5 \times (q_3 - q_1)]$. The dashed lines at the 50% correct response indicator denote the chance level. (B) shows box-and-whisker and scatter plots of the proportions of correct responses for targets 1 and 2 in the training (left) and test (right) trials under the stationary (S), rotating (R), and

chance (C) conditions. As the chance level was at 50%, the rates of correct responses under the chance condition created for GLMMs was assumed to be at 50%. The horizontal lines of the boxes indicate the median. For the chance condition, the horizontal lines indicate the chance level. The bottom and top of the boxes indicate the 25th ($q_1$) and 75th ($q_3$) percentiles, respectively. The bottom and top whiskers extend to the minimum and maximum data points within the range of $[q_1 - 1.5 \times (q_3 - q_1)]$ to $[q_3 + 1.5 \times (q_3 - q_1)]$. The results of the logistic regression based on GLMMs are indicated by asterisks and "ns" (** and *** denote significance at the 1% and 0.1% levels, respectively; ns means non-significant).

2 (maximum: 65.9 ± 0.04 dB; minimum: 57.9 ± 0.06 dB; mean: 61.9 ± 0.04 dB; $N$ = 120 [8 trials × 15 participants]), the difference was small (less than 1.3 dB).

Table 1 summarizes the acoustic cues used for target geometry identification as reported by the participants during the hearing survey. Under the stationary condition, among the 15 participants 14 reported using non-time-varying echo information, 11 of whom reported using timbre and/or pitch cues. Only a few participants (i.e., P9, P10, and P14) reported using other cues, such as the time difference between the pulse and echo, under the stationary condition. By contrast, under the rotating condition, 13 participants reported using time-varying echo information, 12 of whom reported using time-varying patterns in terms of intensity, timbre, and/or pitch.

The participants listened to the pitch-converted binaural echoes from four sides (Fig 3, sides 1 and 2 of targets 1 and 2) in the training trials under the stationary condition. The pitch-converted binaural echoes from the four sides exhibited different energy spectral density (ESD) patterns (Fig 7). This finding suggests that the type of the target (target 1 or 2) and the shape of the side facing the sound source (convex or concave) do not solely determine the similarities in timbre and/or pitch among the pitch-converted binaural echoes. Acoustic similarities among the four pitch-converted binaural echoes in pairs are demonstrated as magnitude-squared coherence (S3 Fig).

## Discussion

### Effective sensing strategies for target geometry identification, as inspired by echolocating bats

In the training and test trials, the participants identified the two target geometries in the presence of time-varying echo information as the acoustic cue (rotating condition), but not without it (stationary condition; Fig 5B, top and middle panels). The results of the hearing survey (Table 1) suggest that sighted people can identify target geometries using time-varying patterns according to echo intensity, timbre, and/or pitch by sensing targets from different angles using buzz-like FM signals. However, performance significantly differed between the stationary and rotating conditions only in the test trials (Fig 5B, bottom panels). This finding suggests that time-varying echo information is effective for target geometry identification by human echolocation especially when echolocators cannot obtain answer feedback during sensing.

Under the stationary condition, although P2, P8, P10, and P15 exhibited extremely low performance (less than 40%) in the test trials (Fig 5A, right panel), P2 and P15 exhibited training performance above the chance level (Fig 5A, left panel). Thus, it appears that under the stationary condition, P2 and P15 can distinguish differences among the echoes but cannot associate the echoes with the target geometries correctly. The type of the target (target 1 or 2) and the shape of the side facing the sound source (convex or concave) do not solely determine the similarities in timbre and/or pitch among the pitch-converted binaural echoes (Figs 7 and S3). This aspect may explain the difficulty of the participants in associating non-time-varying echo information with target geometries—even between the two targets—under the stationary condition. One can easily predict that performance would decline with the use of more target geometries compared with those in the current experiment. These results suggest that non-time-varying echo information is unsuitable for target geometry identification.

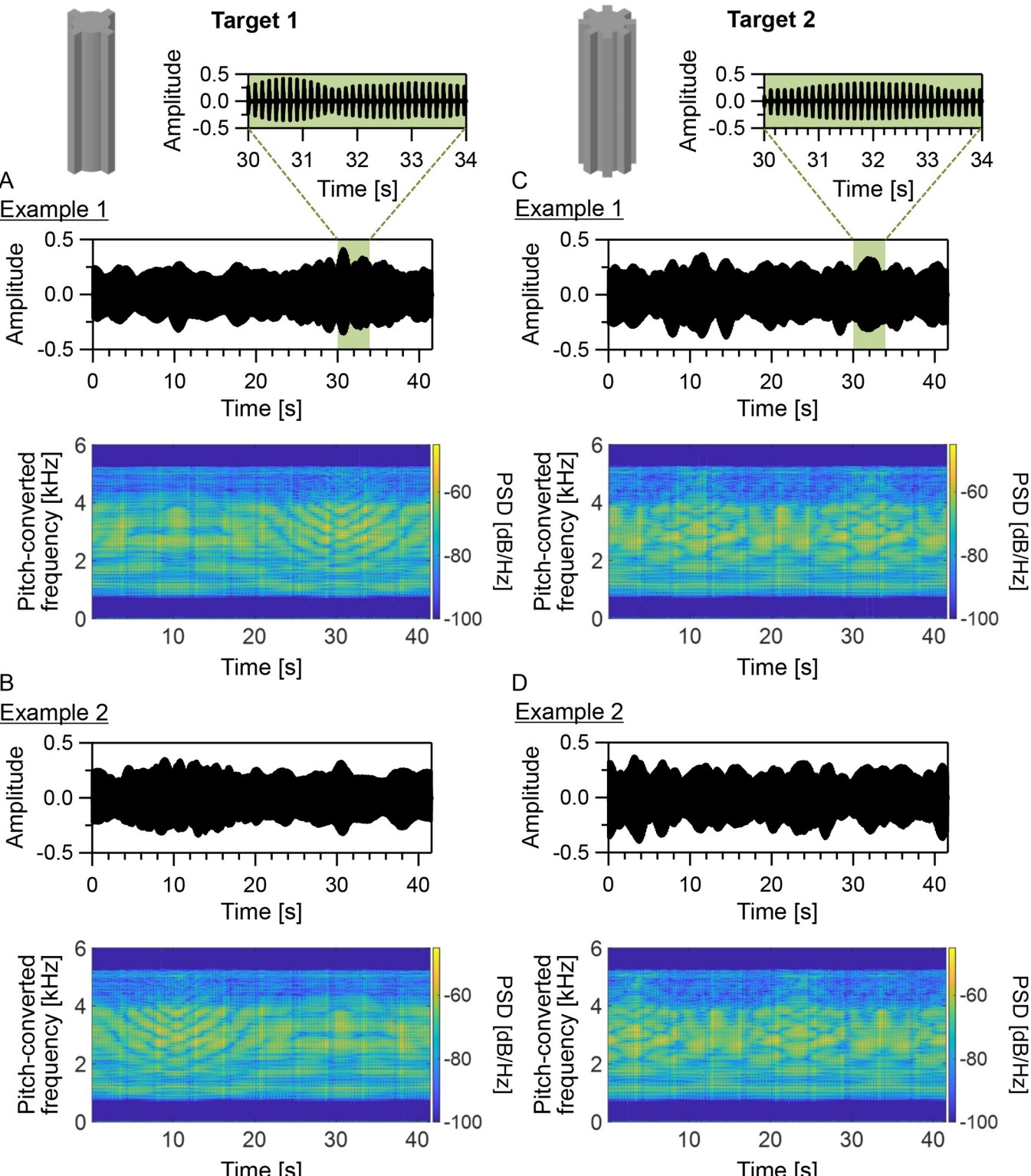

**Fig 6. Time-varying echoes from rotating targets.** (A–D) are examples of amplitude waveforms (top) and spectrograms (bottom) of the pitch-converted binaural sounds to which P12 listened for 41.6 s (5.2 s [signal duration] × 8 [inverse number of the pitch-conversion rate]) in the test trials under the rotating condition. The pitch-converted binaural sounds include the pitch-converted binaural echoes from rotating targets 1 (A and B) and 2 (C and D). The figure also provides examples of enlarged amplitude waveforms (at a timing of 30–34 s; A and C). The spectrograms were calculated using fast Fourier transform (FFT). The sampling frequency was 12 kHz, and the FFT window length was 2,048 points.

**Table 1. Acoustic cues reported by the participants.**

| Participant No. | Stationary condition | | Rotating condition | |
|---|---|---|---|---|
| | Time-varying echo information | Non-time-varying echo information | Time-varying echo information | Non-time-varying echo information |
| P1 | × | Pitch, timbre | Intensity | × |
| P2 | × | Timbre | Intensity, pitch | × |
| P3 | × | Pitch, timbre | Others | Timbre |
| P4 | × | Pitch, others | Intensity, pitch | × |
| P5 | × | Pitch | × | Intensity, timbre |
| P6 | × | Pitch, timbre | Pitch | Timbre |
| P7 | Pitch | × | Intensity, pitch | × |
| P8 | × | Pitch, timbre | Intensity, timbre | × |
| P9 | × | Others | × | Others |
| P10 | × | Others | Intensity | × |
| P11 | × | Pitch | Pitch, timbre | × |
| P12 | × | Pitch, timbre | Pitch | × |
| P13 | × | Pitch | Pitch | × |
| P14 | × | Others | Pitch | × |
| P15 | × | Pitch, others | Pitch | × |

This table presents a summary of the hearing survey results regarding the acoustic cues used by the participants for target geometry identification under the stationary and rotating conditions. The responses are classified as time-varying or non-time-varying echo information, and the specific acoustic features used as the acoustic cues are categorized into four groups: (1) intensity, (2) timbre, (3) pitch, and (4) others. The categorized acoustic features are shown in each cell. Cross marks indicate that the participants did not use the echo information. For example, P1 did not use time-varying echo information as an acoustic cue under the stationary condition.

The study determined the number of training trials (16 trials) based on a restricted experimental duration (up to 2 h). However, this can be increased by conducting the experiment

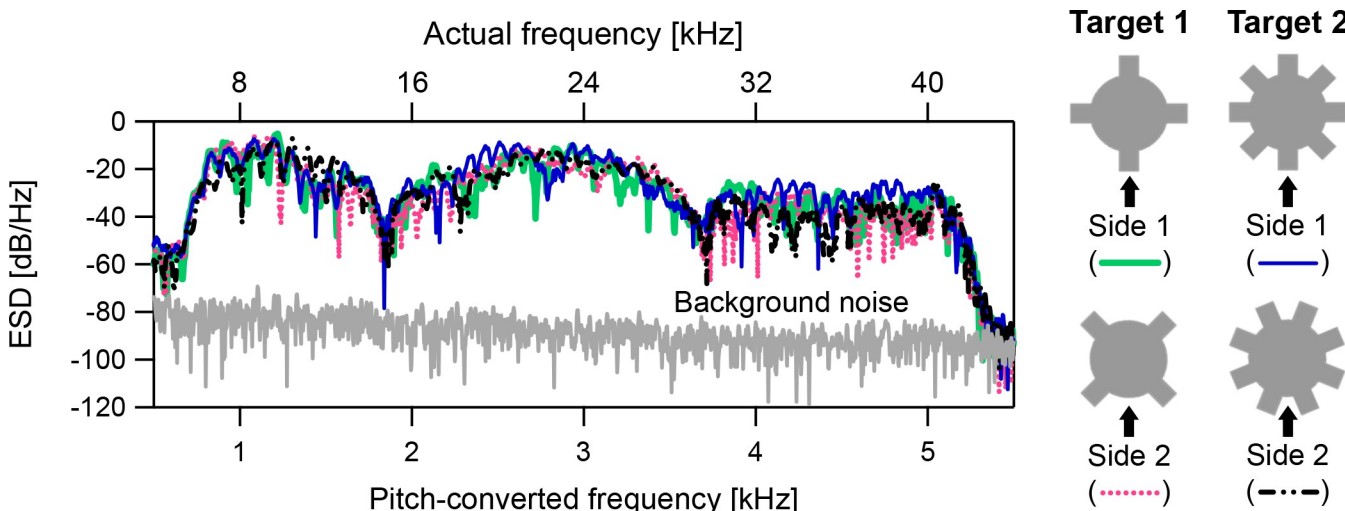

**Fig 7. Acoustic similarities among the pitch-converted binaural echoes from stationary targets.** This figure provides examples of the energy spectral densities (ESDs) of the pitch-converted binaural sounds to which P12 listened in the training trials under the stationary condition. The pitch-converted binaural sounds include the pitch-converted binaural echoes from side 1 of target 1 (light green bold line), side 2 of target 1 (magenta dashed line), side 1 of target 2 (blue line), and side 2 of target 2 (black two-dot chain line). The gray line denotes the background noise floor. Each ESD was calculated by fast Fourier transform (FFT) after averaging 324 readings of the pitch-converted binaural sounds (using the second to the last pitch-converted binaural sounds). The background noise floor was also calculated by FFT after averaging 324 readings of the sounds. The sampling frequency was 12 kHz, and the length of the FFT window was 4,096 points. The top horizontal axis indicates the actual frequency before pitch conversion.

under each condition across several days. Training and test performance may be improved by increasing the frequency of training. However, despite more training, it is possible that several participants will exhibit extremely low performance in the test trials under the stationary condition. For instance, P15, under the stationary condition, displayed extremely low performance in the test trials (18.8%) but not in the training trials (68.8%; Fig 5A). By constantly checking the answer feedback, the participants could familiarize themselves with the associations between the non-time-varying echo information and the target geometries under the stationary condition. By contrast, although the test performances of P13 and P14 under the rotating condition were less than the chance level, no participants exhibited extremely low performance in the test trials (Fig 5A, right panel). These results suggest that time-varying echo information is easier for humans to associate with target geometries and is easier to apply when familiarizing oneself with the associations even if there is no answer feedback.

P4 displayed extremely high performance (81.3%) in the test trials under the rotating condition (Fig 5A, right panel). However, the training performance of P4 was the same as the chance level (Fig 5A, left panel). Thus, P4 may require more time to grasp the technique of associating the time-varying echo information and target geometries, as compared with P1, P11, and P12, who also exhibited extremely high performance in the test trials (Fig 5A). This finding implies that a substantial amount of training may not always be necessary for target geometry identification using time-varying echo information. However, once a person grasps the technique of associating time-varying echo information with target geometries, its application may become robust. This notion may relate to the use of time-varying information for sensing through other sensory modalities (e.g., haptic sense) in daily life.

## Buzz-like signals for acquiring time-varying echo information

The frequency band of the synthetic echolocation signal, which mimics the buzz signals of FM bats, was determined by considering the scales of the targets and MDH as well as the intensity–frequency characteristics of human auditory systems [21]. This mechanism enabled the signal to acquire successfully the remarkable time-varying echo patterns in the spectrograms within its frequency band (Fig 6, bottom panels) and enabled the participants to use time-varying patterns in echo pitch and timbre (Table 1) under the rotating condition. However, one can also easily imagine that the signal cannot acquire time-varying echo information with satisfactory accuracy for target geometry identification under the rotating condition if the IPI and/or duration are unsuitable for the pitch of the convex sides and the rotating speed of the targets. This notion suggests that the appropriate design of the acoustic (e.g., frequency band) and emission (e.g., IPI and signal duration) features of synthetic echolocation signals depends on the specific situation and is important for acquiring time-varying echo information using buzz-like FM signals.

FM bats emit buzz signals at the terminal phase before capturing their prey [2]; this timeframe slightly differs in duration, and is number-dependent, based on the bat species [2] and foraging situation (e.g., foraging in the field and laboratory) [3]. For example, Japanese house bats (*Pipistrellus abramus*) that forage in the field emit approximately 10–30 pulses for 0.05–0.25 s as buzz signals before capturing their prey (the number of pulses and durations of the buzz signals were calculated using the previous field measurement data of the authors in another study [11]). The buzz signals emitted by trained *E. fuscus* approaching a tethered prey in a laboratory were reported to last up to approximately 1 s [3]. Nonetheless, scholars assume that the durations and numbers of buzz signals emitted by FM bats are extremely shorter and smaller than those of the synthetic echolocation signal used in the present study (Fig 2). It is suggested that buzz signals emitted by FM bats prior to capturing their prey are used by bats to

localize and track their prey [25]. Interestingly, longer buzz-like sequences from FM bats (*E. fuscus*) were also observed when the level of difficulty of texture discrimination was high in a texture discrimination experiment [20]. This finding suggests that FM bats use buzz signals to not only localize and track targets for capture but also to obtain time-varying echo information that reflects detailed information of the target, such as texture information. Moreover, FM bats may also adjust the emission features (e.g., duration and number) of buzz signals depending on the purpose of sensing. In this case, the sensing strategies of echolocating bats can be considered by comparing knowledge gained through research on bat biosonar and human echolocation. A similar approach has been proposed in research on dolphin biosonar [22, 26]. Human listening experiments have examined the echo acoustic features used by dolphins for object discrimination, where recording of pitch-converted ultrasonic echoes from dolphin echolocation signals were used and where participants answered questionnaires [22, 26].

## Effect of active sensing on target geometry identification

Echolocating bats recognize the shape and texture of their targets through active sensing using self-produced ultrasounds while flying [19, 20, 27]. Blind echolocation experts can also identify 2D shapes through active sensing using self-produced sounds, such as mouth clicks, while moving their heads [15]. These similarities suggest that echolocating bats and blind echolocation experts can sense the interaction between the motor command information and echo acoustic information in real time during sensing, which may also play an important role in object recognition. However, although the participants in the current experiment could control the time they start sensing independently, they were not allowed to change the sensing positions and pulse directions while listening to the pitch-converted binaural sounds. Therefore, if the participants could sense the interaction between their motor command information and echo acoustic information in real time by simultaneously controlling acoustic sensing and movement, similar to echolocating bats and blind echolocation experts, they may be able to identify target geometries more easily, even if the target geometries are more intricate than those used in the present study. To test this hypothesis, future studies should modify the experimental system, such that the participants can move freely during active sensing and listening to pitch-converted echoes from targets in real time. Binaural sounds recorded using an MDH enabled the participants to obtain 3D sound images outside the head [18]. Therefore, using an MDH in future experiments may be advantageous because 3D sound images may be effectively used for active sensing of targets during movement.

## Supporting information

**S1 Fig. Audiograms of the participants.** Red circles and blue cross marks indicate the hearing levels of the right and left ears of the 15 participants.
(TIF)

**S2 Fig. Target geometry identification performance of the participants in the two experimental orders.** This figure shows box-and-whisker plots of the percentages of correct answers of the participants who participated in the experiment in experimental orders A (P1–P8) and B (P9–P15) for targets 1 and 2 in the test trials under the stationary (left) and rotating (right) conditions. The horizontal lines in the boxes indicate the median. The bottom and top of the boxes indicate the 25th ($q_1$) and 75th ($q_3$) percentiles, respectively. The bottom and top whiskers extend to the minimum and maximum data points within the range of $[q_1 - 1.5 \times (q_3 - q_1)]$ to $[q_3 + 1.5 \times (q_3 - q_1)]$. The result of the logistic regression based on the GLMM is indicated by "ns" (non-significant).
(TIF)

**S3 Fig. Examples of magnitude-squared coherence of the pitch-converted binaural echoes from stationary targets.** (A–F) denote the magnitude-squared coherence corresponding to the acoustic similarities among the pitch-converted binaural sounds to which P12 listened in the training trials under the stationary condition. The pitch-converted binaural sounds include the pitch-converted binaural echoes from sides 1 and 2 of targets 1 and 2. The magnitude-squared coherence was calculated for all six pairs: side 1 of target 1 and side 2 of target 1 (A), side 1 of target 1 and side 1 of target 2 (B), side 1 of target 1 and side 2 of target 2 (C), side 2 of target 1 and side 1 of target 2 (D), side 2 of target 1 and side 2 of target 2 (E), and side 1 of target 2 and side 2 of target 2 (F). The magnitude-squared coherence was calculated using fast Fourier transform (FFT) after averaging 324 readings of the pitch-converted binaural sounds (using the second to the last pitch-converted binaural sounds). The sampling frequency was 12 kHz, and the FFT window length was 4,096 points. The top horizontal logarithmic axes depict the actual frequency before pitch conversion. The middle panels display the superimposed top views of the two targets (light blue and pink targets), and the purple areas indicate the common shapes of the targets. The magnitude-squared coherence suggests that the similarities in timbre and/or pitch among the pitch-converted binaural echoes are not determined only by the type of target (1 or 2) and the shape of the side facing the sound source (convex or concave). (TIF)

**S1 File. Experimental data used for Fig 5A.**
(XLSX)

**S2 File. Experimental data used for Fig 5B.**
(XLSX)

**S3 File. Data used for S1 Fig.**
(XLSX)

## Acknowledgments

We thank Dr. Akiko Callan for sharing the audiometer for purposes of conducting the hearing tests and for the valuable advice on psychoacoustic experiments; Dr. Yoshiki Nagatani for the valuable advice on techniques for acoustic measurements and acoustic analyses; Dr. Olga Heim for the valuable advice on statistical analyses; Kazuki Yoshino and Yumi Fujitsuka for their assistance in the preliminary psychoacoustic and acoustic measurements at the early stage of the study; and Enago, Crimson Interactive Pvt. Ltd. and Editage for English language editing. We also thank the three anonymous reviewers for their valuable comments and helpful suggestions.

## Author Contributions

**Conceptualization:** Miwa Sumiya.

**Formal analysis:** Miwa Sumiya.

**Funding acquisition:** Miwa Sumiya.

**Investigation:** Miwa Sumiya, Kaoru Ashihara, Hiroki Watanabe, Tsutomu Terada, Shizuko Hiryu, Hiroshi Ando.

**Methodology:** Miwa Sumiya, Kaoru Ashihara, Hiroki Watanabe, Tsutomu Terada, Shizuko Hiryu, Hiroshi Ando.

**Project administration:** Miwa Sumiya, Hiroshi Ando.

**Resources:** Miwa Sumiya, Kaoru Ashihara, Shizuko Hiryu, Hiroshi Ando.

**Software:** Miwa Sumiya, Hiroki Watanabe, Tsutomu Terada.

**Supervision:** Hiroshi Ando.

**Validation:** Miwa Sumiya.

**Visualization:** Miwa Sumiya.

**Writing – original draft:** Miwa Sumiya.

**Writing – review & editing:** Miwa Sumiya, Kaoru Ashihara, Hiroki Watanabe, Tsutomu Terada, Shizuko Hiryu, Hiroshi Ando.

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
