## [Decision Letter · Decision Letter 0]

4 Jan 2021

PONE-D-20-35294

Time-varying echo information acquired by bat-inspired ultrasonic sensing is effective for texture identification by human echolocation

PLOS ONE

Dear Dr. Sumiya,

Thank you for submitting your manuscript to PLOS ONE. After careful consideration, we feel that it has merit but does not fully meet PLOS ONE’s publication criteria as it currently stands. Therefore, we invite you to submit a revised version of the manuscript that addresses the points raised during the review process.

We look forward to receiving your revised manuscript.

Kind regards,

Mounya Elhilali

Academic Editor

PLOS ONE

Additional Editor Comments:

This is an interesting study that explores human capability in extracting informative cues from echoes. As noted by the reviewers in details, the authors are encouraged to revise the manuscript and particularly address the issue of subject variability. The graphical displays chosen in the study do not adequately address this issue nor does the paper address the differences between participants and train/test with careful analysis. Furthermore, the authors are advised to revise the write-up as it can be better streamlined for readability and language editing. It contains number of superfluous sections and extremely long and verbose segments.

Reviewers' comments:

Reviewer's Responses to Questions

**Comments to the Author**

1. Is the manuscript technically sound, and do the data support the conclusions?

Reviewer #1: Yes

Reviewer #2: Yes

Reviewer #3: Partly

2. Has the statistical analysis been performed appropriately and rigorously? 

Reviewer #1: Yes

Reviewer #2: Yes

Reviewer #3: I Don't Know

3. Have the authors made all data underlying the findings in their manuscript fully available?

Reviewer #1: No

Reviewer #2: Yes

Reviewer #3: Yes

4. Is the manuscript presented in an intelligible fashion and written in standard English?

Reviewer #1: Yes

Reviewer #2: Yes

Reviewer #3: No

5. Review Comments to the Author

Reviewer #1: The summary data and variance are all available in the supplemental figures, but there is no visualization of raw data points used to arrive at the means. This is easily fixed either by the addition of a figure or a link to the raw data on a hosting service. If the data are unaccessible due to participant anonymity concerns, this should be stated.

Reviewer #2: The authors explore the use of pitch converted echoes from echolocation signals to investigate the acoustic cues that may help identify object identity. The authors describe the use of time-varying information for correct object identification. The experiment is well thought of and interesting. I believe this work is of importance as it may prove to inspire the development of gadgets that may be used by visually impaired individuals for navigation.

I have one major comment regarding the plotting of the data, bar graphs are not the most appropriate way to show this data as the variability between participants fades out this way (a variability which is evidenced in table one by some participants having very high percentages of correct answers, e.g. P1 and P12). This data should be plotted in box plots with all data points visible. I strongly suggest using box plots with all data points visible.

Additionally, I have a few comments that would improve the clarity of the manuscript:

Line 2 - What do the authors mean by appropriately designed ultrasounds? Nothing has been designed in evolution.... Re-word.

Line 8 (and others) - What do the authors mean by “an echolocation signal, including ultrasound”? Anything above 20kHZ is ultrasound. I recommend just referring to “synthetic echolocation signal” and defining it as it is done in the “echolocation signal” part of the methods.

Line 34 - Again, not designed. Re-word.

Line 71 - “acquired multiple sensing”, this sentence needs re-wording, it is unclear.

Line 164 - replace “were not up to” with “were not above”

Line 152 - One graph with the audiograms should be included as supplementary figures.

Line 251 - The section echolocation signal should be brought up to around line 150. In line 123 they only specify an upper limit of 48kHz (line 123). Also at that point it is unclear if it has harmonics of if it is a single FM sweep. These are important parameters that need to be addressed earlier. Bringing the section up would avoid this problem.

Line 256 to 261 - Unclear sentence. How fast does the object rotate? From this sentence I understand that the target rotates at ~0.27 rpm, thus one rotation takes approximately 164 s. If this is correct, this should be stated clearly before this sentence.

Line 270 - When was the feedback presented? Simultaneous to the pitch converted echo? Before? Or after? This is VERY important information in particular for the results in figures S7 and S8.

Line 274 to 276 - Unclear. Place the target where?

Line 292 - “In case the pulses…” re words to “In the case that the pulses…”

Line 368 - Also, figures S4, S5 and S6 should be sub-panels of figure 4 and should also show all data points. It is not only important if there is a difference between stationary and rotating, but one could argue that it is more important to evaluate if the answers are significantly different than chance. Same for figure S7 and S8.

Line 442 - Authors should specify in the ESD is calculated based on an average of several echoes, as background noise may amplify notches in a single echo and thus accentuate differences.

Line 459 - How are the authors differentiating between “correct answer” and mental imagery? This is unclear. If the participants report hearing a specific target rotated how is this different from reporting they imagine a rotating target. Furthermore, since the participants were shown the target before-hand, the imagery does not add any information. This would be much more interesting if the participants were asked to describe what they imagine without having previously seen the targets or told they could either rotate or be stationary. This section of the results doesn’t add much and is confusing. Could be cut.

Line 480 - The authors do not speculate as to why in test conditions without feedback the participants had higher accuracy in the discrimination than in the training conditions. During training they are above chance for the rotating trials but not the stationary trials, yet these two are not significantly different from each other…. Is it because the feedback is present constantly? This should be tackled in the discussion.

Line 506 - Again, it is not clear if the feedback was present all the time during echo presentation or only before or afterwards.

Reviewer #3: The research reported in this manuscript addresses an interesting question: Does the rotation of an ensonified target aid listeners in making echo-based classifications. However, there are serious issues with the results and the presentation of this work that should be addressed in order to make the manuscript suitable for publication:

1. The results show a big difference between training and test trials. What is presented as the main finding of the research, i.e., subjects can make the discrimination only with rotation, exists only

in the test results, but not in the training results. In the transition from training (with feedback) to testing (w/o feedback), the subjects somehow got a lot worse at the stationary targets and quite a bit better for the rotating targets. Hence, the main difference here seems to be due to an interaction between experimental condition (feedback vs. no-feedback, naive vs. experienced) and target condition (stationary vs. rotating) and not due to target condition alone. The results should be presented by stating this clearly throughout the manuscript, i.e., in abstract, results, and discussion.

2. The manuscript presents the task as "texture discrimination". "Texture" is a surface property, e.g., smooth or rough, that refers to geometrical detail on a scale that is much smaller than that of the overall geometry of the object. The distinguishing feature of the two targets used here (number of "gear

teeth") is on the same scale as the target itself. The "gear teeth" are also not small compared to the wavelengths used - at least for the upper portion of the frequency band. Hence, this task should be cast as a "target geometry" discrimination.

3. The manuscript is unnecessarily verbose, e.g., large portions of the discussion are repetitions of the results. In addition, the language is rife with clumsy wordings ("sonar behaviors in biosonar animals"). It would be best to have this looked over by a native speaker.

6. PLOS authors have the option to publish the peer review history of their article (what does this mean?). If published, this will include your full peer review and any attached files.

Reviewer #1: No

Reviewer #2: No

Reviewer #3: No

---

## [Author Response · Author response to Decision Letter 0]

17 Feb 2021

We would like to express our appreciation to the editor and reviewers for taking the time to review our manuscript and for the valuable comments. We have read all of the comments carefully and made an effort to revise the manuscript based on the remarks provided. We hope that the revised manuscript will be acceptable for publication in PLOS ONE.

---

## [Decision Letter · Decision Letter 1]

17 Mar 2021

PONE-D-20-35294R1

Effectivity of time-varying echo information acquired by bat-inspired acoustic sensing in identifying target geometry using human echolocation

PLOS ONE

Dear Dr. Sumiya,

Thank you for submitting your manuscript to PLOS ONE. After careful consideration, we feel that it has merit but does not fully meet PLOS ONE’s publication criteria as it currently stands. Therefore, we invite you to submit a revised version of the manuscript that addresses the points raised during the review process.

We look forward to receiving your revised manuscript.

Kind regards,

Mounya Elhilali

Academic Editor

PLOS ONE

Journal Requirements:

Additional Editor Comments (if provided):

The authors are commanded on their revision of the manuscript. As noted by reviewers, all major edits and comments were addressed. The authors are encouraged to do one more revision of the manuscript focusing on language editing to improve readability.

Reviewers' comments:

Reviewer's Responses to Questions

**Comments to the Author**

1. If the authors have adequately addressed your comments raised in a previous round of review and you feel that this manuscript is now acceptable for publication, you may indicate that here to bypass the “Comments to the Author” section, enter your conflict of interest statement in the “Confidential to Editor” section, and submit your "Accept" recommendation.

Reviewer #1: All comments have been addressed

Reviewer #2: All comments have been addressed

Reviewer #3: All comments have been addressed

2. Is the manuscript technically sound, and do the data support the conclusions?

Reviewer #1: Yes

Reviewer #2: Yes

Reviewer #3: Yes

3. Has the statistical analysis been performed appropriately and rigorously? 

Reviewer #1: Yes

Reviewer #2: Yes

Reviewer #3: I Don't Know

4. Have the authors made all data underlying the findings in their manuscript fully available?

Reviewer #1: Yes

Reviewer #2: Yes

Reviewer #3: Yes

5. Is the manuscript presented in an intelligible fashion and written in standard English?

Reviewer #1: Yes

Reviewer #2: (No Response)

Reviewer #3: No

6. Review Comments to the Author

Reviewer #1: (No Response)

Reviewer #2: All comments have been addressed and I believe this manuscript is a nice contribution to the field. Congratulations!

Reviewer #3: This is a thorough revision that has addressed all my concerns regarding the science aspects of the manuscript. I notice that the English of the manuscript still has very noticeable weaknesses. Hence, I would suggest having a native speaker revise the language prior to acceptance.

7. PLOS authors have the option to publish the peer review history of their article (what does this mean?). If published, this will include your full peer review and any attached files.

Reviewer #1: No

Reviewer #2: No

Reviewer #3: No

---

## [Author Response · Author response to Decision Letter 1]

3 Apr 2021

We would like to express our appreciation to the editor and reviewers for taking the time to review our revised manuscript and for the valuable comments and suggestions. We have read all of the comments carefully and made an effort to revise the manuscript based on the remarks provided. We hope that the revised manuscript will be acceptable for publication in PLOS ONE.

---

## [Editor Report · Decision Letter 2]

8 Apr 2021

Effectiveness of time-varying echo information for target geometry identification in bat-inspired human echolocation

PONE-D-20-35294R2

Dear Dr. Sumiya,

We’re pleased to inform you that your manuscript has been judged scientifically suitable for publication and will be formally accepted for publication once it meets all outstanding technical requirements.

Kind regards,

Mounya Elhilali

Academic Editor

PLOS ONE
---

## [Editor Report · Acceptance letter]

14 Apr 2021

PONE-D-20-35294R2 

Effectiveness of time-varying echo information for target geometry identification in bat-inspired human echolocation 

Dear Dr. Sumiya:

I'm pleased to inform you that your manuscript has been deemed suitable for publication in PLOS ONE. Congratulations! Your manuscript is now with our production department. 

Kind regards, 

on behalf of

Dr. Mounya Elhilali 

Academic Editor

PLOS ONE